# Incidence and Dynamics of CRC Stage Migration: A Regional vs. a National Analysis

**DOI:** 10.3390/cancers16193245

**Published:** 2024-09-24

**Authors:** Carol Faris, Araceli Cuaranta, Michael Abdelmasseh, Rob Finley, Barbara Payne, Alexei Gorka, Juan Sanabria

**Affiliations:** Department of Surgery, Marshall University School of Medicine (MUSOM), Huntington, WV 25701, USA; cuaranta@marshall.edu (A.C.); abdelmasseh@marshall.edu (M.A.); finleyr@marshall.edu (R.F.); payneba@marshall.edu (B.P.); asgorka@gmail.com (A.G.)

**Keywords:** colorectal cancer stages, stage migration, Will Rogers phenomenon, overall survival (OS) by stage

## Abstract

**Simple Summary:**

The purpose of the present study was to determine the role of changes in the stage of colorectal cancer (CRC, Stage 0 to 4, with 0 be very early stage and 4 late stage) at diagnosis over time, and its impact on overall survival (OS) in our Health Network system. Two datasets were used, the Health Network (*n* = 1385) and a national dataset (*n* = 202,391). There was a significant increase in early CRC stages at diagnosis (Stage 0–1), and late diagnosis (Stage 4) with a concomitant decrease in the diagnosis of CRC Stage 2. The present findings confirmed CRC stage migration in our Health Network System, along with a national trend conducive to an increased OS for early CRC stages.

**Abstract:**

Background/Objectives: Due to an increased rate of surveillance colonoscopy, we aim to determine the impact of stage migration on the incidence and overall survival (OS) of patients who underwent pathological staging of colorectal cancer (CRC) at our Health Network System. Methods: Two datasets were included: subjects from the tumor registry at a regional Comprehensive Cancer Center *(n* = 1385) and subjects from the Surveillance, Epidemiology, and End Results (SEER) national database (*n* = 202,391). Results: A significant increase in the diagnosis of CRC Stage 1 and 4 was observed, with a decrease in stage 2, and no change in Stage 3 in the National datasets (*p* < 0.01). There was an increase in Stage 4 CRC diagnosis, with a concurrent decrease in stage 2, and no changes in stages 1 and 3 in the regional dataset (*p* < 0.05). OS followed the expected and progressive decrease in OS by stage (from 1 to 4, *p* < 0.01). Conclusions: The present findings confirmed CRC stage migration in our Health Network System, along with a national trend conducive to an increased OS for early CRC stages.

## 1. Introduction

Colorectal cancer (CRC) remains a global health challenge, accounting for approximately 10% of all cancer cases worldwide [1,2], positioning it as the third most common cancer diagnosed in both men and women [2], and as the second leading cause of cancer-related fatalities [1,2,3,4,5,6,7,8]. Cancer staging guides management of CRC by providing a structure for evaluating disease progression, influencing treatment choices, and forecasting patient prognosis [6,7,8]. The American Joint Committee on Cancer (AJCC) staging framework classifies cancer stages by tumor size (T), lymph node involvement (N), and the presence of metastasis (M) [9,10,11,12,13,14]. Colorectal malignancy stage migration refers to a change in the relative number of cases diagnosed and graded by the TNM system over time. Advancements in diagnostic methods, modifications in staging standards, and enhanced screening initiatives can detect CRC at earlier stages [15,16,17,18,19,20]. This shift can create an impression of improved survival rates without increasing individual patient survival, often depicted as the Will Rogers or lead-time phenomenon. The present study aims to determine the incidence of CRC stage migration, and whether it has a significant possible impact on overall survival (OS) for patients who underwent pathological staging of colorectal cancer (CRC) at our Health Network System.

## 2. Materials and Methods

*Study Design and Setting*: retrospective observational study comparing regional (1988 to 2020) and national trends (1999 to 2022). This study was performed under University IRB approval #2067099-2.

*Data Sources*: for analyses, a regional tumor registry (*n* = 1421, but complete data for *n* = 1385) and the SEER (Surveillance, Epidemiology, and End Results) database (*n* = 202,391), a national cancer resource, were used [21].

*Data Extraction and Selection Criteria*: the timeframe was chosen to capture a comprehensive dataset spanning both the implementation of colonoscopy as a screening tool for the early detection of CRC and allowing for the analysis of long-term trends in CRC stage migration. 

*Inclusion Criteria*:Patients Diagnosed with colorectal cancer (CRC) and included in our tumor registry or SEER datasets.Confirmed CRC by pathology and its TNM staging.

*Exclusion Criteria*:Patients with cancer of the appendix.Unknown CRC TNM stage.

The staging of colorectal cancer (CRC) by the American Joint Committee on Cancer (AJCC) has been a cornerstone in guiding clinical decision making and prognosis. The AJCC staging takes into account the size and extent of the tumor (T), whether cancer cells have spread to nearby lymph nodes (N), and the presence of metastasis (M) [16,22]. From its first edition in 1977 to the current eighth edition, the AJCC staging system for colorectal cancer has incorporated significant changes to reflect a deeper understanding of cancer biology. Earlier editions focused primarily on the anatomical extent of the tumor, whereas recent editions have integrated molecular markers and genetic profiling that influence tumor behavior and, therefore, patient outcomes [10,12,14]. In the latest edition of the AJCC staging manual, CRC staging has been refined to capture the complexity of the disease’s progression. It maintains the established categories from Stage 0 (carcinoma in situ) through to Stage IV (metastatic disease), introducing substages in Stages II and III, which are based on tumor invasion depth (T4a vs. T4b) and the extent of nodal involvement N1 vs. N2. In addition, substages IVA, IVB, and IVC represent the spread of cancer to one distant site, multiple sites, or the peritoneum, respectively [16,22]. These distinctions allow for a more accurate prognosis and targeted treatment approach. To homogenize our patient’s population over the decades of this study, we simplified the staging of the CRC framework into stage 0: carcinoma in situ, stage 1: T1 with no lymph node involvement, stage 2: T2-T3 but has not spread to distant sites or lymph nodes, stage 3: regional lymph node involvement, and stage 4: metastatic disease.

*Statistical Analysis*. All analyses were conducted using R Statistical Software (v4.2.0; R Core Team 2022). For each dataset, monotonic increasing/decreasing trends in the relative proportion of each cancer stage, along with Bowley’s and Kelly’s measures of skewness for summarizing the distribution’s shape, were evaluated using a one-tailed Mann–Kendall test. Significant changes in the relative proportion of cancer stages and the skewness of the distribution establish the occurrence of stage migration. One-tailed Mann–Kendall tests were performed to evaluate trends in the relative proportion of age groups within each cancer stage and examine the evolving age demographic patterns to assess whether stage migration is attributable to advancements in tumor detection methods (trend‘R package, v1.1.6; Pohlert T 2023). Cox regression was subsequently conducted to evaluate the relationship between staging and overall survival (OS), controlling for the year of diagnosis and demographic variables. The significant association between stage and OS, coupled with stage migration, implies changes in survival rates over time. Kaplan–Meier survival curves across various time intervals were then plotted for each cancer stage, and log-rank tests evaluated differences in survival rates among the intervals. Further Cox regressions examined the effect size and direction of year of diagnosis on survival across cancer stages, controlling for demographic variables (survival‘R package, v3.3.1; Therneau T 2022). The significance level for all statistical tests was set at α = 0.05. 

## 3. Results

### 3.1. Trends in Cancer Stage Distribution

In the national database, the analysis shows a significant trend in the distribution of cancer stages at diagnosis over time (Figure 1A). There was an observed increase in the proportion of early-stage (Stage 0 and stage 1, z = 4.075, *p* < 0.001 and z = 3.3623, *p* < 0.001, respectively, by Mann–Kendall test), and late-stage (Stage 3 and stage 4, z = 1.8128, *p* = 0.035 and z = 3.7032, *p* < 0.001, respectively) cases, with a decrease in the proportion of Stage 2 diagnoses (z = −7.6697, *p* < 0.001). In addition, a significant regional trend in the distribution of cancer stages at diagnosis over time was noted (Figure 1B). There was a similar increase in the proportion of Stage 4 diagnoses (z = 1.9608, *p* = 0.025, and before 2020 z = 1.813, *p* = 0.035) with a decrease in Stage 2 diagnoses (z = −1.9624, *p* = 0.025, and before 2020 z = −0.7854, *p* = 0.037).

### 3.2. Survival Analysis

The modeling of overall survival indicates that cancer stage at diagnosis is significantly associated with survival, after controlling for year of diagnosis, age, sex, and race in both databases (national and regional, Figure 2). Stage 1 had 14.6 or 46.9% (national or regional, respectively) higher risk of death compared to Stage 0 (*p* < 0.001); Stage 2 had 106.9 or 61.6% higher risk of death (*p* < 0.001); Stage 3 had 332.3 or 160.9% higher risk of death (*p* < 0.001); and Stage 4 had 2449.1 or 722.5 higher risk of death compared to Stage 0 (*p* < 0.001).

## 4. Discussion

The current state and trends in colorectal cancer (CRC) within the United States show CRC to be the second leading cause of cancer-related deaths nationwide, with an estimated 153,020 new diagnoses/year and 52,550 deaths as of 2023. In addition, the proportion trend in cases (12.7%) and fatalities (7%) occurring in individuals under the age of 50 is a matter of concern [23,24,25,26]. The present study approaches the increase in OS, at least in part from an increase in the diagnoses of early-stages. Our findings support the migration of CRC staging diagnosis, with a significant increase in stages 0 and 1, concurrent with a decrease in Stage 2 at diagnosis. This demographic shift may reflect enhanced screening efforts. In addition, there was a significant increase in the diagnosis of late-stage CRC (stage 4). The increasing trend, particularly among the 30–34 and 30–49 age groups (Figure 3), emphasizes the need for aggressive screening strategies in younger populations. 

Recent data presents a new perspective on CRC incidence and mortality across various age groups, suggesting a demographic shift in the patient population [27]. A decrease in incidence rates is observed solely among individuals aged 65 and older. Since 2011, incidence rates have remained stable among those aged 50–64 but have increased by 2%/year in individuals under 50. This trend is especially evident in the 40–54-year age brackets. Consequently, the younger demographic represents a larger portion of CRC cases, where in 2019 individuals 54 or younger accounted for 20%/year of CRC diagnoses, a significant increase from 11% in 1995. Mortality rates have followed a similar pattern, with a 1% annual increase in individuals younger than 50 and a 0.6% rise in those 50–54 years of age since 2005 [15]. The observed trending incidence served as the foundation for the change in screening policy, where colonoscopy is recommended to start at age 45 rather than 50 [24,28,29]. 

The Will Rogers phenomenon in oncology refers to an apparent improvement in patient outcomes through reclassification into different prognostic groups, recognition of subtle disease manifestations, or earlier disease diagnosis using advanced diagnostic modalities. This phenomenon can lead to “improved” survival statistics without actual improvements in individual patient outcomes due to shifts in the classification of disease stages. It is named after a humorous observation by American comedian Will Rogers, who noted that migration could raise the average intelligence of both the origin and destination locations, analogous to how diagnostic or classification changes can seemingly improve outcomes in both lower and higher disease stages [15]. The observed changes in criteria or techniques result in the reclassification of patients with advanced disease in a lower stage and patients with early disease in a higher stage, thereby improving the statistical outcomes of both groups without a real improvement in patient health [11,14]. Furthermore, according to Märkl et al., evolving trends in the reporting of Stage II and III colorectal cancer across recent years provide crucial insights into the intricate relationship between lymph node count, stage migration, and immunological factors [12]. The quantity of harvested lymph nodes itself holds independent prognostic value, especially in stages II and III of the disease. The conventional explanation for this phenomenon has been stage migration, attributing improved survival rates to more thorough lymph node examinations and the subsequent reclassification of cancer stages based on undetected metastases. In addition, the quality and number of lymph nodes analyzed may not only reflect diagnostic diligence. They could also indicate the patient’s immune response to the cancer, with potentially profound implications for understanding the observed changes in Stage II and III reporting trends [16,30]. 

The significant increase in Stage 0 and Stage 1 diagnoses likely reflects improvements in early detection and screening practices at both national and regional levels. Conversely, the rise in Stage 4 diagnoses suggests that while early detection has improved, there is still a significant proportion of cases being diagnosed at advanced stages. This could be attributed, at least in part, to improved diagnostic imaging technologies, which enable the detection of metastatic disease that might have been missed in earlier decades. Additionally, it highlights potential gaps in screening coverage and access to healthcare services. Factors such as socioeconomic disparities, lack of access to healthcare, and variations in healthcare policy and infrastructure contribute to these trends. Addressing these disparities is crucial for ensuring that the benefits of early detection are equitably distributed across all population groups. Finally, the observed decrease in Stage 2 diagnoses may indicate that cancers previously diagnosed at this stage are now being detected either earlier or later. This shift underscores the importance of continuous advancements in diagnostic methods to accurately classify cancer stages and guide appropriate treatment strategies.

Some differences in the incidence and mortality of CRC have also been observed by sex, race/ethnicity, and geography. The overall annual age-standardized CRC mortality rate, in the most recent five years, stood at 13.1 per 100,000, with mortality rates significantly higher in men (15.7/100,000) than in women (11.0/100,000) [22]. The disease prevalence difference may be attributed to gender-related risk factors or lifestyle. Nevertheless, it highlighted an increased vulnerability of men to CRC, signaling a critical need for gender-specific approaches in prevention, screening, and treatment [22,23,24,25,26]. Furthermore, the interplay of genetic, lifestyle, and socio-economic factors may explain both racial/ethnic and geographic variations in CRC. African Americans have a higher incidence of CRC, 41.7/100,000 [31], second to Alaska natives with a CRC incidence at 88.5/100,000, and a mortality at 35.9/100,000. While certain states exhibit higher rates, West Virginia notably surpasses the national average, presenting an incidence of 50.7/100,000 in men and 38.6/100,000 in women, and mortality rates of 23.9 and 17.0/100,000 for men and women, respectively [27,28,29,32]. 

Other risk factors confer a higher risk for CRC. Familial predispositions stand out, with individuals who have a first-degree relative diagnosed with CRC facing a twofold increased risk; this number is even higher if the family member is diagnosed before the age of 50 [33,34]. Hereditary conditions such as Lynch syndrome (LS) and familial adenomatous polyposis (FAP) markedly elevate risk, with Lynch syndrome carriers facing up to an 80% lifetime risk by the age of 85 [33,35,36,37]. To distinguish LS from the less common familial adenomatous polyposis, the term hereditary nonpolyposis colon cancer (HNPCC) was coined, and for many years the terms Lynch syndrome and HNPCC were used interchangeably. To capture more families with hereditary cancer, the Amsterdam II Criteria was developed. These were based on the same criteria as Amsterdam I but were expanded to include several tumors associated with HNPCC in addition to colorectal cancer. Currently, the term HNPCC is discouraged because these patients do, in fact, have polyps, although not as many as is typical in Lynch syndrome [38].

Lifestyle and other factors further compound CRC risk. Diets rich in red and processed meats can augment CRC risk by approximately 17% and 18%, respectively, for every 100 g and 50 g consumed daily [33]. Conversely, physical inactivity can increase CRC risk by up to 50%, while obesity amplifies risk by 50% in men and 20% in women, compared to in those with an average body mass index. Tobacco smoking and alcohol consumption are equally deleterious; smokers have a two- to three-fold increased risk, and heavy alcohol users (more than three drinks daily) face a 40% higher risk [39]. Individuals with diabetes face a 1.3-fold increase in risk, while those grappling with IBD disease see their risk rise as much as threefold, underscoring the significant impact of chronic illnesses on colorectal cancer susceptibility [33]. Yang et al. revealed the critical role of the gut microbiome in CRC, especially in younger-onset cases [40]. The study identified unique microbial signatures associated with early-onset CRC, including increased diversity and specific bacteria like *Flavonifractor plautii*, which may significantly influence disease risk and progression [41,42,43]. While the study does not quantify risk in fold increases, its findings suggest that microbial dysbiosis could substantially elevate CRC risk in younger populations [40].

Colorectal cancer screening recommendations have evolved significantly since they were first introduced in the 1980s. Initially, screening methods primarily included fecal occult blood testing (FOBT), sigmoidoscopy, and colonoscopy, all aimed at identifying cancer at an early stage in asymptomatic individuals. Over time, as both understanding of the disease and technology improved, additional methods such as DNA stool tests and virtual colonoscopy were introduced, offering more options for early detection [44,45,46]. The U.S. Preventive Services Task Force (USPSTF) advises that adults at average risk for colorectal cancer start regular screening at age 45, a reduction from the previous starting age of 50 [44]. Recommended screening methods include traditional options like high-sensitivity fecal occult blood test FOBT or fecal immunochemical test (FIT) annually, flexible sigmoidoscopy every 5 years, or colonoscopy every 10 years, in addition to newer methods such as the multi-targeted stool DNA test every 3 years [44]. Screening for high-risk individuals starts earlier and is conducted more frequently (Table 1).

Subjects diagnosed with FAP are advised to begin screening as early as age 10–12 with annual or biennial exams until a colectomy is performed. Lynch syndrome carriers should start between 20 and 25 years of age, or 10 years earlier than the youngest affected family member, with a colonoscopy every 1–2 years [44,45]. For those with a personal history of CRC, a colonoscopy is recommended at diagnosis and then at 1-, 3-, and 5-year intervals if no additional abnormalities are found. In inflammatory bowel disease, the risk of CRC increases with the duration and extent of the disease, requiring colonoscopy with biopsies every 1–2 years after 8 years of chronic colitis [44,47].

The nature of our observational study design, which utilizes retrospective data, inherently limits our ability to fully control all possible confounding variables and biases. Any discrepancies or omissions in data entry, coding, or reporting within these registries might have implications for the validity of our study outcomes. Furthermore, our methodology involved the exclusion of patients who did not have clearly defined stages of colorectal cancer (CRC) and those with precancerous polyps. This approach, while necessary for the integrity of our study, could potentially lead to an under-representation of the true incidence rates of early-stage CRC in the population. The lack of detailed information on comorbidities, socioeconomic backgrounds, and lifestyle choices presents a limitation, as these factors are known to have significant impacts on CRC risks and the stage at which the cancer is diagnosed. Finally, no data are available regarding the number of polypectomies that may have impacted CRC development in the registry of the SEER dataset [45]. Each of these limitations must be considered when evaluating our conclusions and the extent to which they can be applied to wider populations and settings.

## 5. Conclusions

Our analysis identified notable shifts in the distribution of CRC stages at diagnosis from 1988 to 2020. Databases showed an increased trend in the proportion of Stage 1 and 4 diagnoses, with a concurrent decrease in Stage 2. This phenomenon aligns with the Will Rogers phenomenon, where advancements in diagnostic technologies and changes in staging criteria result in the reclassification of cancer stages, creating the illusion, at least in part, of improved survival rates without actual changes in patient outcomes. The survival analysis confirms that the stage at diagnosis significantly impacts survival rates. Further research is needed to disentangle the effects of stage migration from advancements in medical treatments on the observed improvements in survival rates.

## Figures and Tables

**Figure 1 cancers-16-03245-f001:**
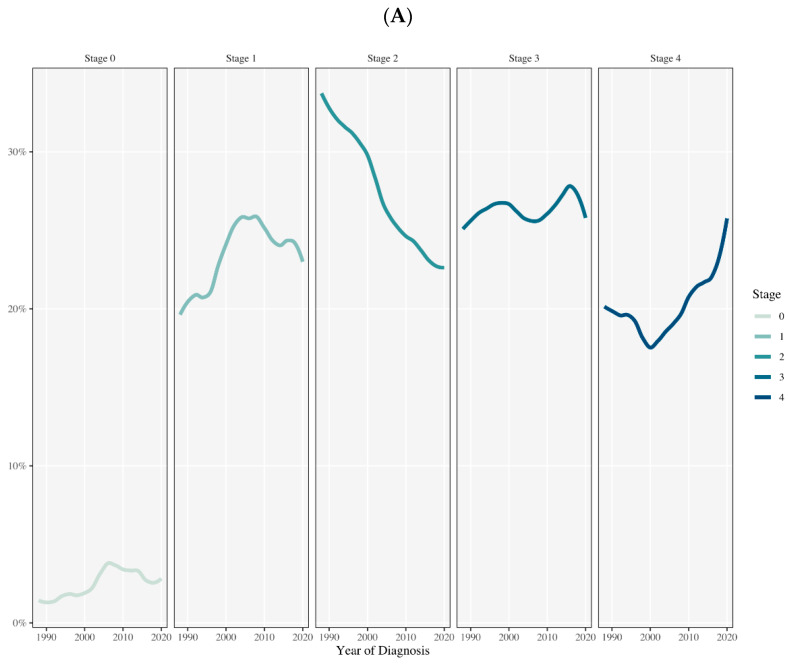
Trends in cancer incidence from Stage 0 to Stage 4 over time based on a national and regional database. The y-axis shows the incidence rate as percentage (%) by stage, and the x-axis shows the year of diagnosis. (**A**) There was a significant increase in the incidence of early (stages 0 and 1) and late stages (Stage 3 and 4), and a significant decrease in Stage 2 diagnosis (*p* < 0.01). (**B**) There was an increase incidence of late stage 4, and a significant decrease in the incidence of Stage 2 diagnoses (*p* < 0.05).

**Figure 2 cancers-16-03245-f002:**
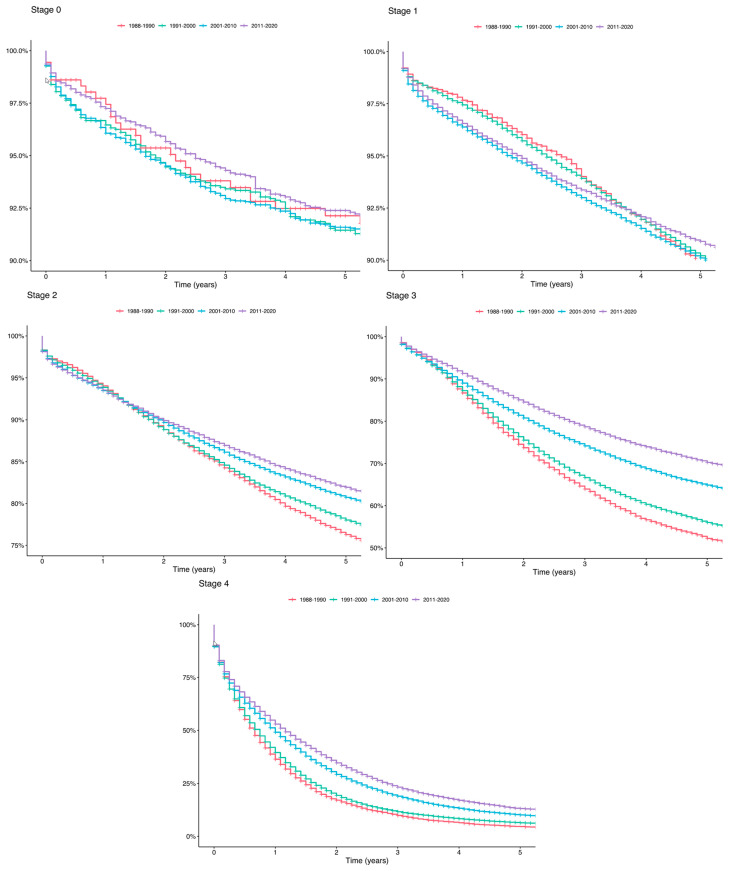
*National overall survival (OS) curves for CRC by stage over time.* The curves represent survival probabilities over a 5-year period. Stage of CRC at diagnosis was associated with OS at 5 years (*p* < 0.01).

**Figure 3 cancers-16-03245-f003:**
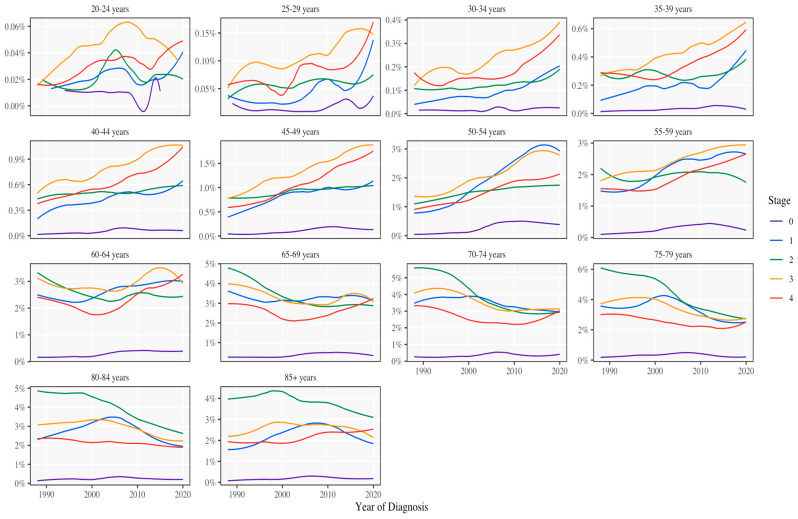
CRC incidence by age from 1990 to 2020. The incidence rates have remained.

**Table 1 cancers-16-03245-t001:** Risk stratification and screening criteria for CRC (2023).

Risk Category	Criteria	Screening Start Age	Screening Test	Interval
Average	Adults aged 45–75 years. No high-risk factors	45 (USPSTF guidelines)	Varies: stool test, colonoscopy etc.	Depending on test used
High	Familial adenomatous polyposis	10–12	Colonoscopy	Every 1–2 years until colectomy
High	Lynch Syndrome	20–25 for 10 years (before the youngest case in family)	Colonoscopy	Every 1–2 years
High	IBD (Crohn’s or UC)	After 8 years of chronic colitis	Colonoscopy with biopsies	Every 1–2 years
High	Family hx of CRC (family member < 60 years or >2 family members)	At age 40 or 10 years earlier than age of youngest family member at diagnosis	Colonoscopy	Every 5 years
High	Family hx of CRC (family member > 60 years)	40	Colonoscopy	Every 10 years
High	Personal hx of CRC	At time of CRC diagnosis	Colonoscopy	1 year after surgery, the at 3 years, and then every 5 years if normal
High	Adult with cystic fibrosis	40	Colonoscopy	Every 5 years

## Data Availability

All data used for statistical analysis is available with no identifiers upon request on excel format, and after confidentiality agreement confirmation and IRB approval.

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
