# Peer review of "Incidence and Dynamics of CRC Stage Migration: A Regional vs. a National Analysis"

_cancers, 2024, doi:10.3390/cancers16193245_

Round 1

Reviewer 1 Report

Comments and Suggestions for Authors

The paper by Carol Faris et al. is a retrospective study of overall survivall of CRC in two cohorts, a local regional comprehensive cancer center (1385 patients) and the Surveillance, Epidemiology, and End Results (SEER) National database (202391 patients) over the periods 1988-2020 (regional) and 1999-2022 (national). The major result was an increase in stage 1 and 4 and decrease in stage 2 CRC over time and an increased OS. The study confirmed a trend of CRC stage migration with more early onset and less late onset CRC,  most likely because of increased screening.

The paper is well written, and clearly describe the material and methods. The results though, are not clearly described. First, figures 1 and 2 are difficult to read  because of smaoll letters and weak colors and second the message could be better described. 

The aim was to determine the impact of stage migration on incidence and overall survival. To understand the impact on incidence and overall survival it would be good to also have overall incidence and survival for all CRC over the period to compare with, to know whether there is an actual increase or decrease in numbers of CRC, and about the total OS in CRC during this period to compare with the OS in each stage.

The most informative figure is no 3 which is in Discussion (and perhaps not based on the data in this study?).  This clearly demonstrates the shift in different ages over time and clearly the rise of all stages before age 50 and a decline after 50. This figure could even be improved by adding a similar box of curves for all ages to illustrate the overall incidence. If this data comes from the National datasets it could serve as a figure no 1.

The current figure 1A is said to show the incidence on the Y-axis and I think it shows the proportion of the total numbers? The curves go up and down a lot over the period which makes them difficult to interpret. The figure 1B looks better, showing curves supporting the interpretations in the text (and figure 3) that early stages (mostly 1) and late (4) increase in proportion and stage 2 declines. Was the figures 1A and 1B named wrong, so 1A was local and 1B national?

Figure 2 shows that current stage diagnostics are still valid to serve as a prognostic tool. This does not really need to be illustrated with the figure 2. Since the aim was to determine the impact on overall survival and therefor it would be good also to know if OS in CRC-all stages has improved. From the figure 2 it looks like OS is improved for stage 2, 3 and perhaps 4 over the period, and perhaps also stage 1 – is this the case? In the abstract it is stated the study showed increased OS for early onset. Since the increase of EO CRC is of particular concern it could be of interest to see this data and a figure showing OS in those before and after 50 in the Results.

In Discussion is brought up possible explanations for a change in CRC incidence and survival besides improved diagnostics. Gender, ethnicity, geographic origin, genetics as well as life-style factors and other environmental risk factors are brought up but could be more interesting if known changes in any of those risk factors are also discussed. 

Minor: 

Headline – Analyses, should be Analysis

Figure 1A does it show CRC incidence on Y-axis, 

Figure 1A and B, since there is one curve for each stage the curves do not need to look different

Figure 3 does not specify what Y-axis is in the text to the figure, only what is on the x-axis.

Author Response

The paper is well-written and clearly describes the material and methods. The results though, are not clearly described. First, figures 1 and 2 are difficult to read because of small letters and weak colors and second the message could be better described.

Response. Thank you for the comment. The figures had been modified, and enlarged, and the figure legend is more descriptive of the message.

• The aim was to determine the impact of stage migration on incidence and overall survival. To understand the impact on incidence and overall survival it would be good to also have overall incidence and survival for all CRC over the period to compare with, to know whether there is an actual increase or decrease in numbers of CRC, and about the total OS in CRC during this period to compare with the OS in each stage.

Response. The reviewer is correct. We have changed our aim to ‘the present study aims to determine the incidence of CRC stage migration, and if it is significant the possible impact on overall survival (OS) for patients who underwent pathological staging of colorectal cancer (CRC) at our Health Network System’.

• The most informative figure is # 3 which is in Discussion (and perhaps not based on the data in this study?). This clearly demonstrates the shift in different ages over time and clearly the rise of all stages before age 50 and a decline after 50. This figure could even be improved by adding a similar box of curves for all ages to illustrate the overall incidence. If this data comes from the National datasets it could serve as Figure # 1.

Response. Figure 3 came from a different dataset, and it is unfortunately not broken by CRC stages.

• The current Figure 1A is said to show the incidence on the Y- axis and I think it shows the proportion of the total numbers? The curves go up and down a lot over the period which makes them difficult to interpret. Figure 1B looks better, showing curves supporting the interpretations in the text (and figure 3) that early stages (mostly 1) and late (4) increase in proportion and stage 2 declines. Was the figures 1A and 1B named wrong, so 1A was local and 1B national?

Response. The figures were re-run by R with a smoother code. They were correct, and they showed in the Y-axis the proportion in the incidence of the CRC stage. This was added to the figure legend to reflect that.

• Figure 2 shows that current-stage diagnostics are still valid to serve as a prognostic tool. This does not really need to be illustrated with figure 2. Since the aim was to determine the impact on overall survival, it would be good also to know if OS in CRC-all stages have improved. From figure 2 it looks like OS is improved for stage 2, 3 and perhaps 4 over the period, and perhaps also stage 1 - is this the case? In the abstract it is stated the study showed increased OS for early onset. Since the increase of EO CRC is of particular concern it could be of interest to see this data and a figure showing OS in those before and after 50 in the Results.

Response. The main aim was to show a migration of the CRC stage, and yes you are correct, the stage tool is still valid as a prognostic tool. The dataset we obtain for the CRC migration has no identifiers including DOB or age at diagnosis. The dataset for CRC incidence by age was provided to us in another dataset with no identifiers, and we do not have a key to merge the two datasets. We plan in another study to see if the biology of CRC in younger patient <50 behaves similarly as the older patients (>50).

• In Discussion is brought up possible explanations for a change in CRC incidence and survival besides improved diagnostics. Gender, ethnicity, geographic origin, genetics as well as life style factors and other environmental risk factors are brought up but could be more interesting if known changes in any of those risk factors are also discussed.

Response. This is a very provocative matter. Nevertheless, as the reviewer may already experience, there are a significant number of publications with positive and negative findings after risk factors modifications mainly by geography, age, gender, and diet. The interactions between environmental factors and genetic influences are complex and exceed the purpose of the present publication.

• Minor: Headline -Analyses, should be Analysis.

Response. Thank you, it has been modified.

Figure 1A does it show CRC incidence on Y-axis.

Response. Thank you, it has been clarified.

Figure 1A and B, since there is one curve for each stage the curves do not need to look different.

Response. The figures curves have been modified.

Figure 3 does not specify what Y-axis is in the text to the figure, only what is on the x-axis.

Response. Figure legend was modified. 

Reviewer 2 Report

Comments and Suggestions for Authors

Please review and comment on each of these questions within the text of your article, complement them or list them as limitations of the study.

1. One major limitation is that the researchers were looking at data that was already collected, so they had no control over how the data was gathered. This can lead to missing information or errors in the data, which might affect the results.

2. Another limitation is that they couldn’t control for all possible confounders, which are other factors that could influence the outcome of the study.

3. The investigators did not adjust for some important confounders, like the patient’s lifestyle, such as diet, physical activity, and smoking, which can affect colorectal cancer risk.

4. They also did not consider socioeconomic factors, which could influence access to healthcare and early detection of cancer.

5. Additionally, they didn't include data on comorbidities (other health conditions) that patients might have, which could also affect their cancer stage and survival.

6. How did you account for differences in colorectal cancer incidence between different geographic regions? Did you consider regional variations in diet, lifestyle, or healthcare access that might influence these trends? The study did not fully account for regional differences in diet, lifestyle, or access to healthcare. These factors can affect cancer incidence and should be considered when comparing regional and national data. It is recommended to include a discussion of these regional variations and how they might impact the study’s findings.

7. No sensitivity analysis was mentioned in the article. Did you use any sensitivity analysis to test how different confounders might affect the results? Without this, how can we be sure the associations you found are robust? This is important to ensure that the results are not overly influenced by specific confounders. The authors should consider conducting and discussing a sensitivity analysis to strengthen their findings.

8. The study did not control for changes in chemotherapy and targeted therapies over time. The study discusses stage migration, but does not mention the impact of advancements in chemotherapy and targeted therapies on survival rates. How did you control for these treatment changes over time? These advancements can significantly impact survival rates and should be discussed as a possible confounder in the results. The authors should include a section on how treatment changes might have influenced the survival outcomes.

9. How did the study account for differences in surgical techniques and outcomes over the study period? Surgical advancements can also affect survival and stage migration. The study did not adjust for changes in surgical techniques, which can affect patient outcomes. The authors should discuss how surgical advancements might have influenced the trends in stage migration and survival and consider this as a potential confounder.

10. The study did not account for advancements in imaging technology that might have improved the accuracy of cancer staging. Were there any improvements in imaging technology during the study period that could have led to more accurate staging of colorectal cancer? How did you account for this? The authors should address this potential bias in the discussion, as better imaging could lead to more early-stage diagnoses and impact survival rates.

11. Did you consider genetic mutations like Lynch syndrome or familial adenomatous polyposis in your analysis? These genetic factors can significantly influence the age of onset and progression of colorectal cancer. The study did not account for genetic mutations like Lynch syndrome or familial adenomatous polyposis, which are important risk factors for early-onset colorectal cancer. The authors should mention these genetic factors and discuss their potential impact on the study’s results.

12. The study did not adjust for dietary factors like red meat and fiber intake, which can significantly influence colorectal cancer risk. How did you account for dietary factors that are known to influence colorectal cancer risk, such as red meat and fiber intake, in your analysis? The authors should discuss how these factors might affect the results and consider them in future analyses.

13. Did you assess the impact of health insurance coverage and access to screening on the stage at which colorectal cancer is diagnosed? These factors can create disparities in early detection.

14. The study did not assess the impact of health insurance coverage and access to screening, which are crucial for early cancer detection. The authors should consider discussing these factors and how they might contribute to disparities in cancer staging and survival outcomes.

15. What cost-benefit analysis was done to evaluate the effectiveness of early screening programs in reducing colorectal cancer mortality? Were the costs of increased screening justified by the survival benefits observed? The article did not perform a cost-benefit analysis of early screening programs. This analysis is important to justify the resources spent on screening. The authors should consider including a discussion on the economic implications of their findings and the potential cost-effectiveness of early screening.

16. The study relies heavily on retrospective data, which can be prone to biases like selection bias and information bias. How did you minimize these biases in your study design and data analysis?

The authors failed to fully adjust the results for these confounders, meaning that the study's findings might not accurately reflect the true relationship between cancer stage migration and survival. This lack of adjustment makes it harder to say for sure whether the observed trends in survival are due to changes in cancer detection and treatment or if they are influenced by other unmeasured factors.

Author Response

The investigators did not adjust for some important confounders, like the patient's lifestyle, such 
as diet, physical activity, and smoking, which can affect colorectal cancer risk. 
They also did not consider socioeconomic factors, which could influence access to healthcare and 
early detection of cancer. Additionally, they didn't include data on comorbidities (other health 
conditions) that patients might have, which could also affect their cancer stage and survival. 
How did you account for differences in colorectal cancer incidence between different geographic 
regions? Did you consider regional variations in diet, lifestyle, or healthcare access that might 
influence these trends? The study did not fully account for regional differences in diet, lifestyle, or 
access to healthcare. These factors can affect cancer incidence and should be considered when 
comparing regional and national data. It is recommended to include a discussion of these regional 
variations and how they might impact the study's findings. 
Did you consider genetic mutations like Lynch syndrome or familial adenomatous polyposis in 
your analysis? These genetic factors can significantly influence the age of onset and progression 
of colorectal cancer. The study did not account for genetic mutations like Lynch syndrome or 
familial adenomatous polyposis, which are important risk factors for early-onset colorectal cancer. 
The authors should mention these genetic factors and discuss their potential impact on the study's 
results. 
The study did not adjust for dietary factors like red meat and fiber intake, which can significantly 
influence colorectal cancer risk. How did you account for dietary factors that are known to 
influence colorectal cancer risk, such as red meat and fiber intake, in your analysis? The authors 
should discuss how these factors might affect the results and consider them in future analyses. 
Did you assess the impact of health insurance coverage and access to screening on the stage at 
which colorectal cancer is diagnosed? These factors can create disparities in early detection. 
The study did not assess the impact of health insurance coverage and access to screening, which 
are crucial for early cancer detection. The authors should consider discussing these factors and  
• how they might contribute to disparities in cancer staging and survival outcomes. 
What cost-benefit analysis was done to evaluate the effectiveness of early screening  
programs in reducing colorectal cancer mortality? Were the costs of increased screening 
justified by the survival benefits observed? The article did not perform a cost-benefit analysis 
of early screening programs. This analysis is important to justify the resources spent on  
screening. The authors should consider including a discussion on the economic implications of 
their findings and the potential cost-effectiveness of early screening.  
The authors failed to fully adjust the results for these confounders, meaning that the study's 
findings might not accurately reflect the true relationship between cancer stage migration and 
survival. This lack of adjustment makes it harder to say for sure whether the observed trends in 
survival are due to changes in cancer detection and treatment or if they are influenced by other 
unmeasured factors.  
The study did not control changes in chemotherapy and targeted therapies over time. The study 
discusses stage migration but does not mention the impact of advancements in chemotherapy and 
targeted therapies on survival rates. How did you control these treatment changes over time? 
These advancements can significantly impact survival rates and should be discussed as a possible 
confounder in the results. The authors should include a section on how treatment changes might 
have influenced the survival outcomes. 
How did the study account for differences in surgical techniques and outcomes over the study 
period? Surgical advancements can also affect survival and stage migration. The study did not 
adjust for changes in surgical techniques, which can affect patient outcomes. The authors should 
discuss how surgical advancements might have influenced the trends in stage migration and 
survival and consider this as a potential confounder. 
The study did not account for advancements in imaging technology that might have improved the 
accuracy of cancer staging. Were there any improvements in imaging technology during the 
study period that could have led to more accurate staging of colorectal cancer? How did you account for this. The authors should address this potential bias in the discussion, as better imaging could lead to more early-stage diagnoses and impact survival rates. 
Response. We have clarified the aim of our study to ‘the present study aims to determine the 
incidence of CRC stage migration, and if it is significant the possible impact on overall survival 
(OS) for patients who underwent pathological staging of colorectal cancer (CRC) at our Health 
Network System’. We also have shown that OS by stage is still a useful prognosticator tool. 
Nevertheless, and although we firmly believe that screening policies and improve imaging in a 
larger population with access to health care had played a significant role in the earlier detection 
and re-staging of CRC (stages 2-3 to stage 4), we cannot offer evidence for the position of any 
factor, and it exceed the purpose of the present study.  

The study relies heavily on retrospective data, which can be prone to biases like selection bias and 
information bias. How did you minimize these biases in your study design and data analysis? 
Response. The reviewer is correct. The nature of retrospective studies implies significant biases as 
sated in our limitations of the study paragraph; “the nature of our observational study design,  
which utilizes retrospective data, inherently limits our ability to fully control all possible  
confounding variables and biases. Any discrepancies or omissions in data entry, coding, or 
reporting within these registries might have implications for the validity of our study outcomes. 
Furthermore, our methodology involved the exclusion of patients who did not have clearly defined 
stages of colorectal cancer (CRC) and those with precancerous polyps. This approach, while  
necessary for the integrity of our study, could potentially lead to an underrepresentation of the 
true incidence rates of early-stage CRC in the population. The lack of detailed information on 
comorbidities, socioeconomic backgrounds, and lifestyle choices presents a limitation, as these 
factors are known to have significant impacts on CRC risks and the stage at which the cancer is  
diagnosed. Finally, no data is available in the registry of SEER dataset, regarding the number of 
polypectomies that may have impacted on CRC development. (64) Each of these limitations must be 
considered when evaluating our conclusions and the extent to which they can be applied to wider 
populations and settings. Nonetheless, our study showed a significant increase proportion in the 
diagnosis of earlier CRC stages, with the expected OS survival by stage. The role of environmental 
and genetic factors and their interactions in the OS, especially in younger subjects (<50) remains 
to be determined.  

• No sensitivity analysis was mentioned in the article. Did you use any sensitivity analysis to test 
how different confounders might affect the results? Without this, how can we be sure the 
associations you found are robust? This is important to ensure that the results are not overly 
influenced by specific confounders. The authors should consider conducting and discussing a 
sensitivity analysis to strengthen their findings. 
Response. One way to see sensitivity analysis, is by testing the robustness of the model examining how it responds to changes, such as excluding certain data, reassigning levels of categorical variables, 
combining levels, adding or removing covariates, modifying assumptions, etc. If a single outlier 
significantly alters the model, the results may be tenuous. We did not conduct a formal sensitivity 
analysis. However, we rigorously tested our results using various exploratory and explicit 
methods. Specifically, we performed univariate, multivariate, and stratification analyses to assess 
the robustness of our findings. These analyses demonstrate that the associations hold across 
different conditions and control for potential biases, ensuring that the results are not driven by 
confounding variables. Beyond that, we could probably improve the estimates for the confidence 
intervals of hazard ratios or Kaplan-Meier survival curves by using bootstrapping. This technique 
involves sampling the data many times with replacement to create a distribution of the sampling 
error. However, given our sample size and the high quality of the SEER database data, we think 
bootstrapping is excessive and unlikely to materially alter our results. 

Round 2

Reviewer 1 Report

Comments and Suggestions for Authors

I still do not understand to read the message in the figure 1A. The changes described in the text of this ms and p-values suggest that the proportion of stage 0, 1, 3 and 4 increase and stage 2 decrease which does make sense to me and is confirmed in the figure 1B (regional) and I would expect it to be even more clear in the national where much more cases would be involved (as was also the case in the calculations of numbers and values). But 1A curves go ups and down with decades. I could understand if they looked like this for a smaller cohort (less sign. values such as for the regional) - to me it looks like figure 1A could be the regional where trends would be less clear because of lower numbers and 1B the national more clearly showing the trends because of higher numbers (much more significant).

Minor:

The authors have addressed my questions and made som modifications but there is still one issue for me regarding the Figure 1.

I understand the figure comes from a program and therefore it was not possible to adjust the size of the text in the figure in relation to the curves. The text says the curves are "proportion of incidence" but figure legend says only "incidence".

Reviewer 2 Report

Comments and Suggestions for Authors

Thank you for comprehensively addressing my comments and implementing the suggested adjustments to your manuscript. Your revisions have enhanced the clarity and rigor of the study. I appreciate your diligent efforts.

Author Response

Thank you very much for your time and support for this study.